Honey bee (Apis mellifera) colony strength and its effects on pollination and yield in highbush blueberries (Vaccinium corymbosum)

Grant Kennedy Judith 1
DeVetter Lisa 2
Melathopoulos Andony Andony.Melathopoulos@oregonstate.edu 3
1 Department of Integrative Biology, Oregon State University , Corvallis , OR , United States of America
2 Department of Horticulture, Washington State University , Mount Vernon , WA , United States of America
3 Department of Horticulture, Oregon State University , Corvallis , OR , United States of America
Gillespie Joseph
Electronic publication date: 2021 Jul 27
Publication date: 2021
Volume: 9
Electronic Location ID: e11634
Received 2021 Feb 11; Accepted 2021 May 27
Copyright: ©2021 Grant et al.
Copyright year: 2021
Copyright holder: Grant et al.
License: This is an open access article distributed under the terms of the Creative Commons Attribution License, which permits unrestricted use, distribution, reproduction and adaptation in any medium and for any purpose provided that it is properly attributed. For attribution, the original author(s), title, publication source (PeerJ) and either DOI or URL of the article must be cited.
License URL: https://creativecommons.org/licenses/by/4.0/

Keywords: Pollination, European foulbrood, Crop pollination, Stocking rate, Cluster count, Floral visitation

Funding: The Oregon Blueberry Commission This work was supported by a research grant from the Oregon Blueberry Commission. The funders had no role in study design, data collection and analysis, decision to publish, or preparation of the manuscript.

==============================
Many pollination studies with honey bees have examined the effect of colony density on crop yield and yet overlook the effect of variation in the population size of these colonies. High colony density in northern highbush blueberry has been met with concerns from beekeepers who feel higher densities will intensify outbreaks of European foulbrood (EFB, Melissococcus plutonius, Truper and dé Clari), a honey bee brood disease. The purpose of this study was to confirm the prevalence of EFB in colonies pollinating blueberries and to determine whether field-level variation in the population of adult workers in colonies explained variation in blueberry fruit set and/or yield. We addressed these objectives over the course of two production seasons at 13 commercial blueberry fields in Oregon, USA, stocked with identical densities of 10 colonies/ha. We confirmed that all colonies had negligible symptoms of EFB at the start of blueberry pollination, but 53% of colonies in 2019 and 41% in 2020 had symptoms immediately following the pollination season. We also validated a method for rapidly assessing adult honey bee colony populations, namely by counting the rate of foragers returning to colonies, and it was found to be strongly correlated to true internal adult bee population independent of year and ambient temperature at the time of evaluation. Using returning forager counts, we determined there was considerable variation in the average population of colonies at each field, ranging from an estimated 10,300 to 30,700 adult worker bees per colony. While average colony strength did not predict variation in fruit set, it was related to variation in yield, independent of year. Our linear model of flight count (as a proxy for colony strength) predicts estimated yield increases of up to 25,000 kg/ha of blueberries could be achieved by colonies stronger than the recommended six frame minimum, suggesting that higher pollination benefits could be achieved without increasing hive density if stronger colonies are promoted.

Introduction

Northern highbush blueberry production in the Pacific Northwest (PNW; Oregon and Washington) has been rapidly expanding with a 21% increase in utilized production of blueberries from 2018 to 2019 (USDA NASS, 2019). In Oregon, 70.5 million kilograms of blueberries were produced alone in 2019, valued at around $134 million and representing ∼23% of total national production (USDA NASS, 2020). Although highbush blueberry cultivars are considered self-fertile, their morphology makes passive self-pollination difficult (Kloet, 1988), resulting in higher fruit set and yields following insect visitation (Delaplane & Mayer, 2000; Free, 1993; McGregor, 1976). Moreover, bee pollination enables cross-pollination among cultivars, which has been shown to result in larger fruit, better fruit quality, and earlier ripening of berries (MacKenzie, 1997; Brewer & Dobson, 1969; Brewer, Dobson & Nelson, 1969). Most of this pollination is the result of visitation by foragers of the managed European honey bee, particularly on large, commercial-scale blueberry production (Isaacs & Kirk, 2010; Gibbs et al., 2016; Hoffman, Lande & Rao, 2018), even though a number of wild bee species have been shown to be more efficient pollinators of highbush blueberry at the individual bee level (Javorek, Mackenzie & Kloet, 2002).

The minimum population size or “strength” of colonies delivered by beekeepers for crop pollination is frequently specified in-state Extension service recommendations (Sagili & Burgett, 2011; Canadian Honey Council & Managing Bees for Pollination, 1995; Delaplane & Mayer, 2000; Frazier, 2016; Mayer & Burgett, 1993). Adult colony population is also regarded as an important factor by growers in judging the quality of rented colonies. In the high-value California almond [Prunus dulcis Miller (DA Webb)]  pollination market, growers who specify minimum colony strength in their contracts with beekeepers were found to pay higher rates for colonies compared to growers who do not specify colony strength (Goodrich & Goodhue, 2020). Yet, despite the considerable attention on colony strength, there have been few studies to date examining the connection between colony strength and crop yields (although see Geslin et al., 2017).

In Oregon and Washington, the minimum colony strength for blueberry pollination is defined as having six Langstroth-sized frames covered with adult bees (Sagili & Burgett, 2011). However, blueberry growers often lack the tools necessary to assess internal frame counts and therefore adult colony populations. Recognizing this limitation, the Extension Service in these states recommends an alternative, non-invasive measure that can be performed by growers, whereby the grower estimates the rate of returning foragers to a colony (Sagili & Burgett, 2011). A colony that is minimally sufficient has more than 100 returning bees per minute at air temperatures 18 °C and above and with winds less than 15 km/hr. Another rapid assessment method developed for California almond pollination, termed “cluster counts”, involves separating colony brood chambers and counting the number of spaces between frames covered by bees (Nasr et al., 1990); However, this method is more difficult for growers to perform without the assistance of a beekeeper compared to assessing the rate of returning foragers. Notably, while cluster counts have been shown to predict adult colony populations (Nasr et al., 1990; Chabert et al., 2021), there are no studies validating the non-invasive method of enumerating the rate of returning foragers method. Furthermore, the returning foragers method is specific to air temperatures that are frequently not achieved during the cool blueberry bloom period in the PNW (DeVetter et al., 2016), which makes the validity of this approach uncertain.

In contrast to the absence of studies assessing the effect of colony population on yield, there have been a number of studies examining the effect of manipulating colony density, also known as stocking rate (Rollin & Garibaldi, 2019). Blueberry growers, on average, use anywhere from 6.2–12.3 colonies per ha, which is considered to be sufficient for pollination services (Brewer & Dobson, 1969; Gibbs et al., 2016; Isaacs & Kirk, 2010). Higher than recommended stocking rates (10 colonies/ha) have also been shown to promote blueberry yield by DeVetter et al. (2016), although notably, none of their fields were stocked with colonies that had an average minimum of 100 returning foragers per minute. However, average air temperatures in this study were largely below the 18 °C threshold described by Sagili & Burgett (2011). Without validation of the returning forager method, it is unclear whether the method underestimates adult populations in colonies, or if colonies delivered for pollination in their study were consistently below grade. The latter possibility raises the question of whether yield could alternatively be increased at a lower stocking rate with stronger colonies.

Increasing stocking rate in blueberries has been met with resistance from some beekeepers due to coincidental reports of a honey bee brood disease seemingly associated with blueberry pollination. European foulbrood (EFB) has been correlated with the pollination of Vaccinium spp. across the US spanning back to the 1980s, although there are no studies documenting this phenomenon in the large blueberry production regions of the PNW (Martin et al., 2019; Gibbs et al., 2016; Wardell, 1983; Lehnert & Shimanuki, 1980; Waite et al., 2003). Oregon beekeepers have reported to one of the authors (A. Melathopoulos) that they are concerned higher stocking rates will result in higher transmission rates of the disease’s causative bacterial pathogen (Melissococcus plutonius, Truper and dé Clari), and lead to more inter-colony competition, resulting in elevated stress that would render colonies even more susceptible to the disease.

The preliminary goal of this study was to determine whether there is an alternative approach to increasing blueberry yield that benefits both beekeepers and growers; not by adding additional colonies per hectare, but by supplying stronger, higher-quality colonies. An additional purpose was to assess whether variation in blueberry yield could be explained by variation in the strength of colonies used for pollination. To address these, we examined the correlation between the internal adult colony population and the rate at which foraging bees returned to the colony to determine if the relationship was strong enough to evaluate apiaries in our study and to provide growers with a reliable assessment tool. Finally, colony brood diseases were evaluated and monitored to verify beekeepers’ experiences with increased brood diseases, particularly EFB, during blueberry pollination. For this last objective, our goal was to document the increase in brood diseases, rather than determine the causes for the increase.

Materials & Methods

Study location

The survey was performed on eight different commercial blueberry operations in 2019 and 2020, located in the Willamette Valley of western Oregon. A single field of the cultivar ‘Duke’ was selected from each operation, although in one operation two ‘Duke’ fields were selected and for four operations a field of ‘Liberty’ was also included. In total, the study included 9 ‘Duke’ and 4 ‘Liberty’ fields. To maintain independence, all fields were distanced a minimum of two kilometers from one another. All field sites were stocked at 10 colonies per hectare, according to guidelines outlined by Oregon State University Extension (Sagili & Burgett, 2011). The blueberry fields in this study were pollinated by a total of five different commercial beekeepers.

Colony strength

Strength was assessed in 2019 (n = 166 colonies) and 2020 (n =161 colonies) by randomly selecting colonies and assessing flight entrance counts (see below). The selected colonies constituted approximately 25% of all colonies pollinating each blueberry field. The counts were conducted once per field at early (∼20% bloom) or peak blueberry bloom. The flight entrance count method of assessing colony strength was validated on a subset of colonies (n = 48) at four different fields in 2019 by comparing flight counts to two additional methods of assessing honey bee colony populations, namely frame-by-frame counts (Liebefeld method) and cluster counts (described below). These multiple assessments were repeated at three fields in 2020 (n = 36), using the same beekeepers and fields as used in 2019. Frame-by-frame and cluster counts were conducted 2-3 times per season, throughout early (20–27 April), middle (3–11 May), and late (15–25 May) bloom. In 2019, each colony was assessed at least twice, first within a week of being placed in fields (early bloom) and second a week before being removed from fields when pollination was complete (late bloom; <10% bloom remaining), while three fields had an additional assessment mid-season in an effort to increase the sample size for the entrance count correlations. In 2020, colony assessments were again conducted twice, once in early and once in late bloom, while one additional beekeeper pollinating blueberries was assessed for colony strength parameters once during mid-bloom. Notably, this additional beekeeper was not located in a field used for any measures of flower visitation, fruit set or yield (see section below). Colony entrance count assessments were conducted within 48 h of the same colonies being assessed for cluster counts or frame-by-frame assessments.

The first method we used for assessing colony population is known as the Liebefeld method, which involves assessing each Langstroth-sized frame in the colony for coverage with adult bees and sealed brood. This labor-intensive method represents the current standard for assessing honey bee colony populations (Dainat et al., 2020). We compared populations assessed using the Liebefeld method to two, labor-saving methods. Cluster counts were assessed as described by Nasr et al. (1990), which involved counting the number of frame-spaces occupied by bees from the tops and bottoms of frames without removing frames from the box. Cluster count assessments were performed immediately prior to assessments using the Liebefeld method and when temperatures were below 15 °C and before foragers departed at the beginning of the day. The second alternative method was the flight entrance count method described by Sagili & Burgett (2011), and involved counting the number of returning forager bees returning to colonies over a 1-minute period between 10:00 AM to 4:00 PM when temperatures were above 12 °C, there was no precipitation, and winds were less than 15 km/hr. Although Sagili & Burgett’s (2011) recommendations specify ≥ 18 °C in their assessments, temperatures at or above this threshold were seldom achieved and an adjusted threshold of 12 °C was recorded for this study, which reflects natural field conditions (Tuell & Isaacs, 2010). Counts were made from video recordings of flight activity of all colony entrances. Videos were slowed down post-production, and the number of honey bees that completely entered the colony within the time interval was manually counted.

Colony diseases

All colonies evaluated by frame-by-frame population assessments (see above) were also evaluated for the following honey bee brood diseases: EFB (M. plutonius), American foulbrood (caused by Paenibacillus larvae bacteria), chalkbrood (caused by Ascosphaera apis fungus), and sacbrood (virus). Assessments were conducted at the onset of pollination and immediately before they were moved out of the fields in 2019 and 2020. Field diagnosis of the brood diseases was performed by an experienced honey bee pathologist (A. Melathopoulos). EFB was confirmed using molecular techniques described by Wood et al. (2020). Briefly, whenever EFB larvae were identified in an apiary on a given survey date, a minimum of three EFB-diseased larvae were collected using a sterile swab, crushed and stored at −20 °C after returning to the lab at the end of the day. The samples were shipped frozen to the University of Saskatchewan where they were cultured on KSBHI agar. Any colonies resembling M. plutonius were subcultured on KSBHI and confirmed using gram staining and duplex PCR. Confirmed M. plutonius were further characterized using multi-locus sequence typing (MLST) to detect atypical genetic variants of the disease. The number of diseased larvae/pupae per colony was scored using an 3-point ordinal scale adapted from Spivak & Reuter (2001) where 0 = no disease, 1 = 1 − 10 diseased cells per colony, 2 = 11 − 50 diseased cells per colony, 3 >51 diseased cells per colony.

Flower visitation, fruit set, berry weight, and yield

Honey bee visitation measurements outlined in this section were adapted from Courcelles, Button & Elle (2013), with methods following DeVetter et al. (2016). Measurements were conducted on 30 randomly selected blueberry bushes per field in both 2019 and 2020. Bushes were located within 3 randomly selected rows within each field, with 10 bushes randomly selected per row per (10 plants per transect ×3 transects per site) (DeVetter et al., 2016). Honey bee and bumble bee (Bombus sp.) visitation assessments were conducted twice (early and peak bloom; ∼20 and 100% bloom, respectively) and involved counting the total number of legitimate pollination events over a 1 min observation period for each bush for each period (i.e., honey bee foraged within the flower and entered through the corolla rather than nectar robbing). Floral visitation was conducted between 10:00 AM to 4:00 PM when temperatures were at 12 °C or above and there was no precipitation and less than 15 km/hr wind. Fruit set was estimated by counting the total number of flowers and fruit (at the stage when fruit was ∼4 mm in diameter) on four randomly selected flowering clusters per tagged bush. The average number of flower clusters per cane and number of canes per tagged bush were also recorded. Immediately prior to commercial harvest (∼75% fruit had turned blue) 10 berries were randomly picked per bush (i.e., 300 berries per field) and weighed in order to determine the average berry weight. Estimated yield per field was calculated with the following formula: yieldacrelbs=avgberryweight×avgberriescluster×avgclusterscane×avgcanesbush×bushes∗acre

*bushes/acre was provided as an industry standard of 2000.

Statistical analysis

Data were analyzed using Program R (R Core Team; R Foundation for Statistical Computing, Vienna, Austria). Differences in the average strength, as well as the severity of EFB from the beginning to the end of blueberry pollination, were compared with two-sample Wilcoxon sum rank tests. Hives that only had one assessment were not included in these comparisons. We used Kendall’s Tau correlation coefficient to assess whether returning flight counts were a comparable colony strength metric alongside cluster counts and could be a reliable indicator of adult bee populations in the field, as assumptions for the Pearson correlation of normal distribution of data were not consistently met for all variables. Kendall’s Tau correlation coefficients were also implemented to determine whether there were any significant relationships between this new estimate of colony population (mean return rate of foragers), estimates of pollination activity (mean honey bee flower visits, mean bumble bee flower visits), and field-level yield parameters (mean fruit set, mean flowers per bush, mean berry weight. For the former assessment, any hive that was not assessed using all three measures or whose measurements were taken longer than 48 h apart were dropped from the final yield correlation analysis. We tested the hypothesis that returning flight counts predict true adult bee populations using least square means linear regression, with year of study as a factor and ambient temperature at the time of the returning forager count as a covariate. We also used least square means linear regression to test the hypothesis that flight counts predicted yield, with year of study and cultivar as factors, confirming the assumptions of the model by quantile plots and scatterplots of residuals versus predicted values.

Results

Colony quality during blueberry pollination

Colonies did not experience significant growth during the 3–4 weeks of blueberry pollination in 2019 (Fig. 1; W = 914.5, df = 42, P = 0.93). Moreover, colonies experienced elevated numbers of brood diseases, with 63% being free of symptoms before pollination and 19% remaining free of diseases by the end of pollination, with colonies having an average of 1 distinct brood disease. The growth in the incidence of EFB between in-field placement and when they were removed was distinctive in both 2019 and 2020 (Fig. 2; 2019 - W = 495, df = 85, P < 0.001; 2020 - W = 219.5, df = 63, P < 0.001). Levels of EFB in colonies before pollination were very low in both years, with only a single beginning colony in 2020 showing symptoms at the lowest disease rating (Fig. 2B). Molecular analysis of samples of EFB-diseased larvae analyzed by the University of Saskatchewan confirmed that some of the strains found in colonies were of a strain of M. plutonius that is atypical and causes high levels of larval mortality (personal correspondence, Sarah Woods). In 2019, very low levels of both chalkbrood and sacbrood were observed in early and late bloom, with neither having significant increases throughout the season. Sacbrood had a slight increase in severity with colonies having an average rating of 0.1 in early bloom and 0.7 in late bloom, however, the magnitude of this change was not comparable to that which was seen in EFB. In 2020, minor increases in sacbrood prevalence and severity were again observed, while chalkbrood and AFB cases were virtually undetectable.

Figure 1 Average colony strength before and after pollination.

A comparison of average colony strength (measured in frames of adult bees) at four different commercials farms before and after pollination. The horizontal line in each box signifies the median field strength for all farms, boundaries exist at the 25th and 75th percentiles, and whiskers extend to the most extreme data point that is no more than 1.5 times the length of the box. Data were collected in northwest Oregon in 2019.

Figure 2 EFB severity before and after pollination.

The relative severity rating (0–3) of European foulbrood (EFB) found in individual colonies located at four blueberry fields, as colonies enter (Early) and leave (Late) pollination in (A) 2019 and (B) 2020. A rating of 0 indicated that no EFB was detected at the colony, and a rating of 3 was the most severe case a colony could be assigned (see text for description). Dots indicate the rating assigned to an individual colony at the time of the assessment. One farm from 2019 was no longer available in the 2020 assessment, but the other three remain consistent across the two years. Data were collected in northwest Oregon.

Validation of flight entrance estimates of colony strength

Incoming flight entrance counts (mean 68.86; SD ± 31.94; range 11–144) were significantly correlated to both the current standard method of assessing adult bee populations using frame-by-frame coverage (Liebefeld method) and cluster counts (Fig. 3) (P < 0.001). Notably, flight entrance counts were also significantly correlated to colony sealed brood population. Incoming flight entrance counts were significantly related to adult populations estimated using frame-by-frame counts (R2 = 0.37, P < 0.001), independent of the year of assessment (P = 0.6) or variation in-flight temperature (P = 0.3) across a temperature range from 13−23 °C (Fig. 4). We estimated that for every 13 bees returning to the colony per minute, there 2,400 additional adult bees in the colony adult bees, assuming  2,400 bees per Langstroth-sized frame (Burgett & Burikam, 1985).

Figure 3 Comparison of colony strength assessments.

Correlations among the adult and brood population (Liebefeld method) and two labor-saving assessment methods, cluster counts (cluster) and returning flight counts (flight) (see text for description, n = 49 colonies). Correlation coefficients (Kendall’s tau) are listed in the lower cells and designated in strength by color as indicated by the scale. Ellipses indicate: (1) correlations with P-value < 0.05 (i.e., cells without an ellipse are ≥0.05) and (2) the color and eccentricity of the ellipse, which is scaled to the correlation value according to Friendly (2002).

Figure 4 Colony strength linear model.

The relationship between the frames of adult bees in a colony to the rate of returning foragers at its entrance (n = 84 colonies). Each dot represents a colony, and the color of the dots indicates the temperature at which the returning flight data were recorded. Our linear model prediction for the number of frames of adult bees (frames) depends on the number of ‘x’ returning bees to the colony (R2 = 0.32, P < 0.0001). The trendline approximates that for every thirteen returning bees, the internal frame count of adult bees is predicted to increase by one, with a 95% confidence interval. The horizontal dotted line indicates the 6 minimum recommended frames of adult bees for adequate pollination as recommended by Sagili & Burgett (2011).

Colony quality and yield

Average colony strength per field estimated from flight entrance counts was a better predictor of yield compared to floral visits on bushes in both years, and notably, in 2019, colony strength was slightly negatively correlated with honey bee flower visits and bumble bee flower visits (Fig. 5). Only 2% of bush visitation surveys in 2019 and 5% in 2020 recorded one or more bumble bee visits, the highest count from a single visit being 13 in 2019. No other bee taxa were observed visiting flowers during the study. Overall, variation in colony strength and the number of flowers per bush appeared most strongly associated with variation in yield. There was a significant positive fit between the average colony strength and yield in both 2019 and 2020 (R2 = 0.42, P < 0.001), independent of the year (P = 0.5) and blueberry cultivar (P = 0.3) (Fig. 6). We estimate that for every 10 bees per minute increase in average colony population, growers would realize a 4450 kg/ha increase in blueberry yield.

Figure 5 Comparisons among fields and estimated yield.

Correlations among various field components in (A) 2019 (n = 12 fields) and (B) 2020 (n = 13 fields). From top to bottom: honey bee flower visitation, bumble bee flower visitation, flight of returning foragers at colony entrance, percentage of blueberry fruit set, number of flowers per field, average berry weight in kg, and yield approximation in kg/ha. Positive correlation coefficients (Kendall’s tau) are displayed in blue and negative correlations in red as indicated by the legend of the right. Ellipses indicate: (1) correlations with p-value < 0.05 (i.e., cells without an ellipse are ≥0.05) and (2) the color and eccentricity of the ellipse, which is scaled to the correlation value according to Friendly (2002).

Figure 6 Colony strength as a predictor of yield linear model.

The relationship between the estimated field-level blueberry yield in relation to a predictor of average field colony strength, namely the average rate of returning foragers to colonies in the same field (n = 25 fields, 2019 and 2020 data combined). Each symbol represents a field, and the shape and color of the symbol indicate the variety planted at each field. Our linear model prediction for estimated blueberry yield (yield estimate) again depends on the number of ‘x’ returning bees to the colony (R2 = 0.421, P = 0.0008). The trendline approximates that for every 23 returning bees, the estimated yield is predicted to increase by 10,000 kg/ha, with a 95% confidence interval. The vertical line indicates the 6 minimum recommended frames of adult bees for adequate pollination as recommended by Sagili & Burgett (2011) translated into 38 bees per minute using the equation provided in Fig. 4.

Discussion

Our study is the first to demonstrate that yield benefits associated with crop pollination are related to returning forager rates as a proxy for the strength of honey bee colonies brought into pollination. We observed a 70,000 kg/ha difference in estimated blueberry yield among fields in our study stocked with 10 colonies/ha, and predicted that around 40% of this variation could be explained by variation in returning foragers as an indicator for honey bee colony strength. By creating a linear model where incoming flight entrance counts predict estimated blueberry yield, we saw a 62.6% drop in the estimates of yield between fields stocked with colonies at the 25th percentile of strength (6.4 adult frames; 42.6 bees per minute) and the 75th percentile (10.2 adult frames; 90.3 bees per minute). Moreover, the non-invasive method we used to assess honey bee colony strength (flight entrance counts) should be easy for growers to perform and are correlated with more labor-intensive methods used by growers and crop consultants in other cropping systems. We were able to refine the estimates of adult colony population previously developed by Sagili & Burgett (2011), who estimated that a colony meeting the minimum colony strength pollination standard of 6 frames of adult bees should have 100 returning foragers per minute. These findings suggest the returning forager count for 6 frames of adult bees should be 2.6 times lower than previously reported, or 38 returning foragers per minute, and that air temperature does not influence this relationship. Finally, we confirmed reports from Oregon beekeepers that colonies do not experience significant growth during blueberry pollination and large increases in brood diseases, particularly EFB, occur. Combined our research suggests that growers could realize a substantial yield increase by maintaining 10 colony/ha stocking rates, but selecting beekeepers providing stronger colonies for pollination or adjusting colony density based on colony strength. Furthermore, these results may encourage more blueberry growers to establish contracts with their beekeepers, as frequently done in almond production, to help ensure quality.

It is still unknown to what extent EFB is caused by blueberry pollination, or whether a confounding variable is leading to its association with the blueberry pollination season. Explanations for the EFB phenomenon in blueberries include: a nutritional deficiency with blueberry pollen (Girard, Chagnon & Fournier, 2012), low surrounding floral diversity (Colwell et al., 2017), poor weather conditions during the pollination period (Tuell & Isaacs, 2010), pesticide use (Wood et al., 2020), and the alkalinity of the pollen itself (Wardell, 1983), but so far none have been isolated as the sole or primary determinant. While our study determined the presence and prevalence of EFB in a recommended stocking rate, our approach was descriptive and, as such, unable to test the contention of beekeepers that the disease is more severe at a higher stocking rate. It is quite possible, for example, that the disease increases in early spring independent of stocking rate or whether colonies are placed in blueberries. There remains a need for research that investigates the relationship between colony density and the severity of EFB during blueberry pollination. At minimum, there should be a confirmation of the causal connection of EFB to blueberry, perhaps by comparing disease occurrence among colonies from commercial beekeeping operations that are randomly allocated to blueberry and to another unrelated but coincidentally blooming crops such as pear or sweet cherry.

A key limitation of this study is that while Arrington & De Vetter (2018) randomized the stocking rate to fields, findings in this study are correlational and depended on variation within commercial production systems. Future studies should control for colony strength by randomly assigning strong and weak colonies to different fields. A powerful future experimental design could employ a 2 ×2 factorial exploring the interaction between stocking rate and colony strength, providing growers with additional insight into balancing these two factors off when making decisions around how many colonies to stock.

Both this study and Arrington & De Vetter (2018) found that flower visitation did not predict yield like other studies suggest (Isaacs & Kirk, 2010). These results are different from those observed on lowbush blueberry (Aras, De Oliveira & Savoie, 1996) and rabbiteye blueberry (Vaccinium virgatum var. ‘Climax’) (Dedej & Delaplane, 2003; Danka, Sampson & Villa, 2019). One explanation for this discrepancy is that by performing a single assessment of flower visitation per field in a given day, we failed to fully account for flower visitation, which can have strong diurnal patterns in blueberries (Drummond, 2016). In contrast, hive entrance counts remained predictive of yield regardless of annual variation and differences in ambient temperatures. While entrance counts themselves are not labor-intensive to survey, the process of reviewing video footage of colonies took as long as 2–3 min per colony to assess, especially at higher returning forager counts where slow-motion video replay was required to achieve an accurate estimate. What is not clear from our research is whether reduced accuracy associated with visually estimating forager return rate would be sufficient to discriminate among broad categories of colony population strength. For example, future research may focus on the capacity of growers and crop consultants to discriminate among colonies in the median, lower, or upper quartile of returning forager counts from our study (i.e., 25th quartile = 42.6 bees/minute, predicted frame count = 6.4; median = 62.6 bees/minute, predicted frame count = 8; 75th quartile = 90.3 bees/minute, predicted frame count = 10.2), which corresponded with significant increases in yield. Based on the experience of the author responsible for transcribing the returning forager count videos (K. Grant), it seems likely that a trained grower would be able to differentiate between such broad categories. Artificial intelligent algorithms have also been suggested as an even more efficient and similarly accurate process, but price and availability would limit their range (Reka, 2016; Magnier et al., 2018; Kulyukin, 2017).

What is also unclear from our study is whether some of the variation in the ability to predict frame counts from colony flight is driven by parameters such as weather conditions from the preceding day (Clarke & Robert, 2018) or colony state parameters such as the amount of pollen and nectar stores in colonies (Dogterom & Winston, 1999). Better predictive models might result from multivariate analyses of factors other than colony population that influence colony forager entrance counts.

Arrington & De Vetter (2018) found that when doubling the stocking rate from 10 colonies/ha to 20 colonies/ha, that there was a 22% and 40% increase in estimated yield for each of two years in their study, respectively. Colonies used in their study were also evaluated for strength using the returning forager method and were comparably stronger (117 returning foragers per minute) with no differences between treatments. Our results suggest comparable yield increases achieved while doubling stocking rate from 10 to 20 colonies/ha could be achieved through increasing the average population of bees per field. For example, a field stocked with 10 colonies/ha with an average of 38 returning bees per minute (the six frame minimum, Fig. 6), could achieve a 22% increase in yield if colonies were strengthened to a point where the return rate was 54 bees per minute. Similarly, placing colonies with flight counts from the 50th percentile of our survey (62.6 bees per minute), could lead to a 35% increase in yield compared to colonies meeting the six frame minimum. Notably, the difference in estimated yield between the field with the weakest colonies in our study (average count of 11 bees per minute) to the strongest (119.3 bees per minute), suggests that the former field could yield a hypothetical 250% increase if stocked with the strongest instead of the weakest colonies. While other horticultural variables can influence total yield, results from this study suggest that stocking blueberry fields with fewer, stronger colonies should be less expensive than stocking at higher densities assuming colony costs are the same and yields benefits are maintained.

This study draws into question the concept of a minimum strength honey bee colony, as highlighted in numerous crop pollination guides, as our research suggests a linear increase in yield based on colony population. For example, Sagili & Burgett (2011) state that the colonies delivered for blueberry pollination should have a minimum of 6 frames covered with adult bees. This standard is also outlined in the Oregon Administrative Rules (Chapter 603, section 55-005, filed with the Secretary of State on August 17, 1960, as Administrative Order AD 643). Upon further research, we found that these guidelines have since been repealed (Personal correspondence, Ben Zeiner, Oregon State Archives) and at the time of this publication, there is no legal basis in Oregon regarding minimum colony strength for pollination, although the division title still references the “Oregon Standards of Bee Colony Strength” (OAR, 2020). Rather than minimum colony population, our findings suggest the potential to develop a rental pay schedule, whereby growers pay more for the average population of colonies delivered to pollination.

Our findings could potentially provide both blueberry growers and beekeepers with higher economic returns. Blueberry pollination is one of the highest users of honey bee colonies for pollination in the PNW region, using approximately 10,000 colonies (rentals valued at $528 thousand dollars in 2016) (Sagili & Caron, 2017). Blueberries are also the most valuable honey bee-pollinated crop in the state, with a value of utilized production estimated at $134 million in 2019 (USDA NASS, 2020). If the average price per pound of blueberries is $2.20/kg, the difference in gross returns between fields with populous (ex. eight frames; 62.6 bees per minute; the observed median) versus weak (under six frames, ex. four frames; 12.5 bees per minute) colonies would be ∼$49,000/ha, according to both Figs. 4 and 6. Given that colonies for blueberry pollination rent, on average, for $543/ha, it is possible that blueberry growers who incentivize the delivery of stronger colonies could realize more than 100% returns on investment for beekeepers delivering premium strength colonies, even when factoring in a modest rental fee increase for those colonies. For beekeepers, if higher pollination benefits can be achieved without increasing hive density, this may also lower their operating costs if it resulted in lower levels of EFB, although it remains unclear if stocking density or blueberry pollination cause the observed increases in EFB. The correlation between colony strength and the rate of return of bees from the field will allow growers to assess colony strength after colonies are delivered and potentially adjust fee schedules based on their estimates of colony strength.

Conclusions

We hypothesized that target yield could be achieved by increasing the strength (i.e., forager population) of honey bee colonies supplied to blueberry pollination without increasing colony density. We validated and refined existing colony population metrics and extrapolated the relationship between adult frame counts and the rate of returning foragers. We found that average colony strength per blueberry field, estimated using the returning forager counts, predicted yield. We also confirmed that honey bee brood experienced elevated levels of disease, most notably EFB, post-blueberry pollination, although our study was descriptive and unable to attribute the increase in disease to blueberry pollination. Colony strength was found to have a positive influence on estimated blueberry yield, suggesting similar profits to increasing colony density could be achieved. Future directions for research include measuring the effect colony density has on colony strength and EFB severity, economic studies that calculate the cost-benefit trade-offs of supplying stronger colonies for pollination, and identifying whether growers can visually estimate flight counts accurately or whether they require technological assistance.

Supplemental Information

Supplemental Information 1 R-code for data analysis

Information on code included inline in the file.

Click here for additional data file.

Supplemental Information 2 Datasets on flight activity at colonies, bee visitation on crop and berry production

This also includes honey bee strength and disease assessment.

Click here for additional data file.

We would like to thank Jason Myer, Tom Peerbolt, and their team at Peerbolt Crop Management for their help and cooperation with this study, without which this project would not have been possible. We are also extremely grateful for the beekeepers and growers who agreed to let us evaluate their colonies and fields each year.

Additional Information and Declarations

Competing Interests

Author Contributions

Data Availability

The authors declare there are no competing interests.

Kennedy Judith Grant conceived and designed the experiments, performed the experiments, analyzed the data, prepared figures and/or tables, authored or reviewed drafts of the paper, and approved the final draft.

Lisa DeVetter conceived and designed the experiments, authored or reviewed drafts of the paper, and approved the final draft.

Andony Melathopoulos conceived and designed the experiments, performed the experiments, authored or reviewed drafts of the paper, and approved the final draft.

The following information was supplied regarding data availability:

The raw data, including a metadata tab explaining the variables, and R-code for conducting analysis for this study are available in the Supplemental Files.

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
