# Peer review of "Honey bee (Apis mellifera) colony strength and its effects on pollination and yield in highbush blueberries (Vaccinium corymbosum)"

_PeerJ, doi:10.7717/peerj.11634_

## Round 0.1 · original submission · Major Revisions

Dear Dr. Grant and colleagues:

Thanks for submitting your manuscript to PeerJ. I have now received three independent reviews of your work, and as you will see, the reviewers raised some concerns about the research (mostly the manuscript). Despite this, these reviewers are optimistic about your work and the potential impact it will have on research studying honey bee colony strength as it pertains to effective blueberry pollination. Thus, I encourage you to revise your manuscript, accordingly, taking into account all of the concerns raised by all three reviewers.

Please improve the clarity of your manuscript (many suggestions raised by the reviewers). Please also ensure that your figures and tables contain all of the information that is necessary to support your findings and observations. Revise incorrect information.

Please address concerns regarding pathogens. Avoid speculating in the absence of sound controls. Key literature needs to be included to best place your work within the larger context of the field. All work must be repeatable, so provide all necessary information either in the Materials and Methods or supplemental material.

I look forward to seeing your revision, and thanks again for submitting your work to PeerJ.

Good luck with your revision,

-joe

Reviewer 1 ·

Basic reporting

There are some clarity issues in text and minor changes that could be done to improve figure presentation. Please see comments for authors below for details.

Raw data: items listed in metadata do not match the data sheets in the supplemental excel file?

Experimental design

There is no control treatment for brood disease evaluation. This might be OK for the purpose of reporting the observed change over time but not to be presented as an effect of hive density or blueberry pollination.

Validity of the findings

Please check if your data meet the assumptions for Pearson's correlation analysis. Some of the count-based data may be more compatible with non-parametric alternatives.

Additional comments

This paper presents a study in which the authors investigated whether similar benefits of employing more hives to pollinate blueberries could be achieved by increasing colony strength instead of increasing hive density. The authors also compared three methods of evaluating colony strength (frame area coverage, cluster size determined by frame counts, and returning foragers at hive entrance) and concluded that the least invasive method could reliably predict colony strength. Lastly, the authors reported increased EFB infestation in colonies at the end of the blueberry pollination season.
The manuscript for most part is well written with clear language although it feels less polished toward the end.
The experimental design is sound for an observational study. I wish a “control” component had been included to the EFB evaluation, for example, hives placed at a lower density (to test if the increase infestation is associated with density) or hives not placed in blueberry fields. Absence of the control treatment is not a deal breaker given the primary objective was about hive strength and yield, and the difficulty of conducting large scale field experiments with honey bees. However, the authors need to be very careful about how they interpret the results. What you found was an increased level of EFB in most colonies after a period of time. Without a control treatment, you cannot conclude that the increase is related to aggregation of hives or blueberry pollination. It could simply be the phenology of the disease as well but there is no way to distinguish these causes in this study.
Selection of the statistical analysis— please check the underlying assumptions of Pearson’s correlation analysis to make sure the dataset meets the requirements. Otherwise use nonparametric alternatives.
Other minor issues:
Several brood diseases were checked during hive inspections, according to Methods. Why only EFB data is reported? Although you’ve set monitoring EFB as one of the objectives, data on the other diseases would be equally valuable. I would suggest summarizing the findings in text and/or including the data in the supplemental file.
There are some ambiguities that could benefit from additional clarification. Please see line-by-line comments below.

Line 40: Data or statistics of the 45,000 kg/ha increase is not presented in results? Only an estimate of 4450 kg/ha increase for every 10 bees/min is presented (L258). Then in Discussion you say there is a 70,000 kg/ha difference?
L 77-79: Please include citation for the recommendation of counting returning bees.
L 96-97: Arrington & DeVetter 2018 reports an average of 117 returning bees/minute in randomly selected colonies in their study. But here you say none of the fields were stock with colonies with an average minimum 100 returning bees/min. Did I miss something?
L97: If “this study” refers to Arrington & DeVetter 2018, their bee count data were recorded at air temperature above 18 degree C, according to their methods.
L98-100: at a quick glance, the message I get from the Arrington & DeVetter 2018 paper is that the colony foraging strength was above the 100 bees/min threshold and did not significantly differ between their treatments of 10 vs. 20 hive/ha, although the raw data of colony weight were not presented. So they did validate colony strength in their study. It sounds odd to speculate that the colonies in the cited study were consistently below grade without evidence. Perhaps the authors have access to the unpublished data but readers would not know that. Either way, I think this paragraph could be revised to reflect the citation more accurately.
L125-127: It seems redundant to state again the three main objectives that you already explain clearly in this paragraph. Also, you’ve only mentioned the assessment of EFB, which is only one of the many aspects of honey bee health. Saying you are assessing “the effects of blueberry pollination on the health of honey bees” is an exaggeration. “Define honey bee colony quality and strength parameters” does not make much sense as a sentence. I suggest deleting these sentences in L125-127.
L137: Spell out OSU at the first mention.
L140-141: are n =166 and 161 the total number of colonies stocked in fields (so ~40 colonies were assessed each year) or the number of colonies (of the 25%) assessed? Please clarify.
L148: Instead of “2-3 times per season, report the exact dates or information on the length of interval between assessments.
L148-151: Conflicting information— in L148 you say frame-by-frame and cluster counts were done 2-3 times per season but then you say the assessment was done twice in 2019 and only once in 2020. Could you clarify?
L168: just a thought related to the threshold temperature – has there been any studies on correlations between colony strength (bee population) and flight activities in cold weather? I wonder if a strong colony is better at maintaining its cluster temperature than a weak colony, and therefore allows workers to start foraging at a lower ambient temperature threshold? Perhaps this would further emphasis the importance of supplying strong colonies for pollination?

L190: Why only honey bees and bumble bees were recorded? Do other bees visit blueberries?
L209: Why did the analysis only compare EFB, but not the other diseases you’ve also recorded (AFB, chalkbrood, and sacbrood)?
L211-216: some of the parameters are not continuous data (e.g., cluster counts and bee counts). A non-parametric comparison such as Spearman’s rank-order or Kendall’s Tau may be more appropriate for these types of data.
L238: Provide descriptive statistics for incoming flight counts (e.g. means, range etc.)
L246 See comment for Figure 4 below.
Discussion:
L276, L279-280: Could the increase in EFB infestation over time be explained by the phenology of the disease? In early spring honey bee colonies typically do not produce much brood, and therefore the absence of the brood disease symptoms could mean (1) absence of the pathogen in the colony or (2) the bacterial pathogen is still in the spore form due to a lack of brood that allows them to propagate. Ideally the experiment would include a control group of bee colonies that were not moved to blueberry fields, and demonstrated that the control hives had less EFB. Since there is no such control in this study, I feel the authors should at least discuss alternative explanations as well, and not fixate on the conclusion of hive aggregation in for crop pollination as the reason of increased EFB detection.
After reading the paragraph about EFB (L346-356), I would suggest deleting the sentence in L280 about reducing EFB as a potential benefit of reducing colony density and keep the EFB discussion in one place. The single statement at the beginning of the discussion could be misleading.
L291-295. This is a great discussion on predicting yield benefits based on number of returning bees at hive entrances, and would be a great tool for growers to evaluate whether their rental colonies meets their pollination target. However, beekeepers at large still use frame counts to determined colony strength quality during inspection before delivering the hives to crop fields. Could you include a brief discussion on how the bee counts convert to frame numbers in respect of the projected yield benefits?
The final Discussion paragraph on the financial aspects belongs together with the paragraphs that calculates yield increase (L284-301) and hive rental policies (L311-323) to improve the flow of content.

L377. The first sentence (colony strength increases blueberry yield) doesn’t make sense. I suggest “target yield could be achieved by increasing the strength of honey bee colonies without increasing colony density”.

L384: Delete “without aggravating brood diseases”. Your experiments did not test if the prevalence of brood diseases was related to hive density.

Figure 1: Change the Early box and whiskers to dark color lines. Present both Early vs. late in the same color unless there is a reason to use different colors.

Figure 3. I think presenting scatter plots with P values might be more informative than the ellipses alone. Also, check to make sure your date meet the linearity assumption for Pearson’s correlations.
Caption: Repetition of “adult (adult)” and “brood (brood)” is unnecessary. Specify that these are frame measurements from the Liebefeld method.

Figure 4: Rotate the figure. Present the equation as y = 0.8x + 3.
I think the trend line interpretation of every 13 bees = 1 increase in frames of adult bees is quite clever. This would probably be better perceived than the very 10 bees = 3.8 frame in text (L245-247) because a whole frame is easier for a reader/beekeeper to visualize than fractions of frames.

Figure 5. Again, I think it would be more informative to also present the scatter plots.

Figure 6. Rotate figure and present equation as y = 445x + 14427

Supplemental Materials:
- Some of the datasets listed in Metadata are apparently missing: total, alldata, diseasesplus2020, alldata2020, alldatabothyears, total2020
- What is the difference between flight and newflight?
- Does the “Disease” column present the number of different diseases recorded?
- What are the measurement units for Adult (e.g. 14.25) and Brood (e.g 2.25)?

Reviewer 2 ·

Basic reporting

see below

Experimental design

see below

Validity of the findings

see below

Additional comments

General comments
This study investigated the blueberry pollination system with honey bees, in particular how colony strength impacts yield in the crop as well as disease (European foulbrood) in the colonies. Beekeepers have been noticing an uptick in disease when pollinating blueberry, and the hypothesis is that the stocking density may be too high leading to facilitated pathogen spread. The study fixed stocking density at 10 hives per hectare, but measured the variation in the colony strength by quantifying nest variables (bees and brood) as well as entrance flight counts (bees per minute). They found that EFB did indeed significantly increase in both years, and that colony strength varied significantly and was overall fairly low. They also found that yield was positively associated with colony strength, thus increasing colony strength rather than stocking density may help mitigate disease spread while increasing yield.
Overall this is a nice study that constitutes quite a bit of work. My first read was a bit less generous, thinking that there was some circular reasoning in their hypothesis (stocking density and colony strength are both means to increase forager population) but in the end I think it’s a fair enough question that was able to be teased apart. My main criticism is how the pathogen confirmation results were largely dismissed, especially given how EFB can be difficult to diagnose in the field and overlaps symptomatically with several other conditions. Nonetheless, I believe that this study nicely adds to the literature in this area and helps reinforce some useful recommendations for both growers and beekeepers.


Specific comments
1. Line 22-23: perhaps move “crop” from early in the sentence to just before “yield’ to make it more clear that the subject is the crop and not the colony? Also, consider changing “increasing” to “high” since the stocking density was actually not varied in this study (I understand that the densities are have been increased historically but that is out of context in the abstract).
2. Lines 58-59: most wild bee species have been shown to be more efficient pollinators than honey bees ‘at the individual bee level,’ so please add the latter clause since native bee populations cannot be easily manipulated.
3. Lines 69-70: I found this hard to believe, and even a quick Google search found several papers (The impact of honey bee colony quality on crop yield and farmers' profit in apples and pears; Geslin et al. (2017) AGRICULTURE ECOSYSTEMS & ENVIRONMENT; Comparison of Honey Bee (Hymenoptera: Apidae) Colony Units of Different Sizes as Pollinators of Hybrid Seed Canola; Ovinge & Hoover (2018) JOURNAL OF ECONOMIC ENTOMOLOGY). Change to “few” and avoid unnecessary hyperbole.
4. Lines 90-92: there is a really nice recent meta-analysis on this topic that should be cited here: Rollin, O., & Garibaldi, L. A. (2019). Impacts of honeybee density on crop yield: A meta-analysis. Journal of Applied Ecology, 56(5), 1152-1163. doi: 10.1111/1365-2664.13355
5. Lines 110-114: stocking rates, increased disease transmission, and competitive stress are all confounding variables that can lead to increased EFB rates. There is also another factor, exposure to fungicides, that is not discussed here but is highly pertinent to this system.
6. Lines 177-180: would be better to briefly describe the molecular methods in Wood et al. 2020, including sampling and handling procedures in getting the samples to the lab. EFB is a particularly problematic disease to diagnose from symptoms, so it might be good to articulate those as well. Since one of the main findings from this study centers around these diagnoses, it is surprising that there is only one sentence in the methods devoted to their detection.
7. Lines 195: here it states that floral visitation measures were taken less than 15km/hr wind (also stated lines 166 during entrance counts) but in line 79 is states 4.5 m/sec when referring to the recommendations. These appear to be close but not equal (my calculation is 16.2km/hr), so you might wish to keep these units consistent.
8. Lines 216-218: I understand the reasoning for dropping colonies that were not measured within 2 days from each other (although that is fairly conservative), but I don’t understand the reasons for dropping colonies without full data. If these were included, how would the correlations differ?
9. Line 237: It really would be better to report the actual pathology data than cite a personal correspondence. Both the methods and the results on the laboratory confirmations seem quite dismissive to me.
10. Figure 5: theoretically, one would expect a non-linear relationship here since the proportion of foragers increases as colony population increases, but it looks like there’s more scatter in the data to tease out that nuance. Might be interesting to at least test to see if it’s a better fit, but not a big deal if not found in my opinion.

Reviewer 3 ·

Basic reporting

As detailed below in Comments there are omissions to literature context that must be addressed. There's also logic thread issues that fall under "basic reporting" shortfalls.

Experimental design

As detailed below, I have a question about "year" as a covariate.

Validity of the findings

As detailed below, your data and analyses are sound, but you need to be more careful and contextualize your paper with the larger literature.

Additional comments

PeerJ Review of Grant et al. 2021. Honey bee (Apis mellifera) colony strength and its effects on pollination and yield in highbush blueberries (Vaccinium corymbosum)

1. Line 82 “perform” misspelt
2. Lines 133, 134 and elsewhere, use of ‘Varietal’ name apostrophes inconsistent
3. Line 211 sentence beginning “We used Pearson correlation . . .” makes “variation in true adult and sealed brood populations” the grammatic subject, then proceeds to list it along with dependent clauses (1), (2), etc. thus confusing subject with predicate.
4. Lines 214-215 related to above: you need a (3) in front of “estimates of pollination activity . . .” and a (4) in front of “estimates of colony population.”
5. Line 213 no “and” in front of (2)
6. However, I would elevate your putative (4) predicate line 215 “estimates of colony population” to the subject of the sentence! As it stands, here and throughout the paper to this point there is a confusion of cause and effect. Essentially you want to say (1) entrance flight activity is (2) a reliable indicator of adult bee population which is (3) positively associated with bee field density which is (4) positively associated with bee flower visitation rate which (5) improves harvest metrics. This logic thread is vague and muddled.
7. In fact this problematic sentence discussed in #s 3-6 strikes at my central critique of this paper. I’m puzzled why the authors have ignored, evaded, or understated the obvious – that colony population, whether measured by Liebefeled, Nasr, or their clever entrance flight metric, is simply a proxy for the biologically meaningful effect – bee flower visitation rate. From the abstract through the introduction, this central relationship isn’t even mentioned until methods on lines 185. The authors ignore papers like Aras et al. 1996, Dedej et al. 2003 and Danka et al. 2019 that clearly show this relationship, while foregrounded distractions like “hive density” which is probably least important of all. I strongly urge the authors to position their paper into this larger biological context, and clarify exactly what they are showing.
8. Line 221 “the” misspelt
9. Line 219 as year is a categorical, not continuous effect, I don't think it qualifies as a covariate; a block? A plot? Please clarify this language and what you did.
10. Line 226 Finally demonstrating a relationship between EFB and blueberry pollination is very useful.
11. As is your hive entrance population estimator.
12. Lines 262-263 here and through the Discussion and Conclusions be clear that “strength translates to visitation translates to yield.” Be careful of leaping from 1 to 3!

---

## Round 0.2 · accepted · Accept

Dear Dr. Grant and colleagues:

Thanks for revising your manuscript based on the concerns raised by the reviewers. I now believe that your manuscript is suitable for publication. Congratulations! I look forward to seeing this work in print, and I anticipate it being an important resource for groups studying honey bee colony strength as it pertains to effective blueberry pollination. Thanks again for choosing PeerJ to publish such important work.

Best,

-joe

Reviewer 2 ·

Basic reporting

Satisfactory.

Experimental design

Satisfactory.

Validity of the findings

Satisfactory.